# The RNA-Binding Protein HuD Regulates Alternative Splicing and Alternative Polyadenylation in the Mouse Neocortex

**DOI:** 10.3390/molecules26102836

**Published:** 2021-05-11

**Authors:** Rebecca M. Sena, Jeffery L. Twiss, Amy S. Gardiner, Michela Dell’Orco, David N. Linsenbardt, Nora I. Perrone-Bizzozero

**Affiliations:** 1Department Neurosciences, University of New Mexico School of Medicine, Albuquerque, NM 87131, USA; RmSena@salud.unm.edu (R.M.S.); AGardiner@salud.unm.edu (A.S.G.); MichelaDellOrco@salud.unm.edu (M.D.); 2Department Biological Sciences, University of South Carolina, Columbia, SC 29208, USA; TWISS@mailbox.sc.edu; 3Department Cell Biology and Physiology, University of New Mexico School of Medicine, Albuquerque, NM 87131, USA

**Keywords:** HuD, *Elavl4* KO, alternative splicing, alternative polyadenylation, neocortex

## Abstract

The neuronal Hu/ELAV-like proteins HuB, HuC and HuD are a class of RNA-binding proteins that are crucial for proper development and maintenance of the nervous system. These proteins bind to AU-rich elements (AREs) in the untranslated regions (3′-UTRs) of target mRNAs regulating mRNA stability, transport and translation. In addition to these cytoplasmic functions, Hu proteins have been implicated in alternative splicing and alternative polyadenylation in the nucleus. The purpose of this study was to identify transcriptome-wide effects of HuD deletion on both of these nuclear events using RNA sequencing data obtained from the neocortex of *Elavl4^–/–^* (HuD KO) mice. HuD KO affected alternative splicing of 310 genes, including 17 validated HuD targets such as *Cbx3, Cspp1, Snap25* and *Gria2*. In addition, deletion of HuD affected polyadenylation of 53 genes, with the majority of significantly altered mRNAs shifting towards usage of proximal polyadenylation signals (PAS), resulting in shorter 3′-UTRs. None of these genes overlapped with those showing alternative splicing events. Overall, HuD KO had a greater effect on alternative splicing than polyadenylation, with many of the affected genes implicated in several neuronal functions and neuropsychiatric disorders.

## 1. Introduction

RNA-binding proteins (RBPs) serve important functions in the co-transcriptional and post-transcriptional control of gene expression [1,2]. In particular, RBPs play a prominent role in alternative splicing and alternative polyadenylation in the nucleus and in mRNA stabilization, transport and localization in the cytoplasm. 

It has been estimated that 92–95% of human genes undergo alternative splicing to produce multiple protein isoforms from one gene [3,4]. These events occur at the highest frequencies in the brain, with neuron development and function particularly impacted [5]. Alternatively spliced proteins are involved in synaptogenesis, neuronal migration, axon guidance and synaptic transmission and plasticity, all of which have been linked to RBP control in the brain [6]. Neurodegenerative, neurodevelopmental and neuropsychiatric disorders have also been associated with alternative splicing [7,8,9]. Therefore, alternative splicing must be tightly regulated in the nervous system.

The function of mRNAs is also greatly influenced by the 3′-untranslated region (UTR). The length of the 3′-UTR is determined by the selection of a poly(A) site (PAS), and approximately 70% of known human genes have multiple PAS [10]. In the brain, a large proportion of mRNA diversity is attributed to polyadenylation, and a large number of transcripts in the nervous system possess longer 3′-UTRs [11,12,13]. These longer 3′-UTR isoforms mediate mRNA localization to dendrites and axons and can also be subjected to RBP and microRNA-mediated decay in the cell [12]. For example, alternative polyadenylation of the *Bdnf* mRNA regulates translation of the BDNF protein in response to neuronal activity [14]. It has been reported that *Bdnf’s* short 3′-UTR isoforms are restricted to soma, while long 3′-UTR isoforms localize to dendrites [15]. Thus, alternative polyadenylation is an important regulator of mRNA diversity and function in the nervous system.

The Hu proteins HuR, HuB, HuC and HuD constitute a family of RBPs that are mammalian homologs of *Drosophila* ELAV [16,17]. While HuR is ubiquitously expressed, HuB, HuC and HuD, also called neuronal ELAV-like proteins (nELAVs), are expressed primarily in neurons and required for neuronal development, maturation and synaptic plasticity [18,19,20]. Hu proteins are also known to stabilize mRNAs by binding to AU-rich elements (AREs) in the 3′-UTR and poly(A) tails [21,22,23]. 

Nuclear ELAV has been shown to coregulate alternative splicing and polyadenylation in the *Dscam1*, *ewg,* and *fne* genes [24,25,26,27]. In addition, HuB, HuC and HuD have been implicated in the regulation of alternative splicing and alternative polyadenylation in neurons [28]. Specifically, nELAVs promote neuronal calcitonin gene regulatory peptide (CGRP) expression by blocking inclusion of the calcitonin exon in pre-mRNA [29]. Hu proteins can also bind to the proximal PAS of this gene to shift towards distal PAS usage [30]. Furthermore, nELAVs drive the production of short protein isoforms of neurofibromatosis type 1 (NF1) through exclusion of exon 23a [31]. The mechanism by which these RBPs control alternative splicing is mediated by an RNA-dependent increase in localized histone hyperacetylation [32]. 

The role of Hu proteins in nuclear and cytoplasmic events has been previously examined using high-throughput crosslinking as well as HuC/HuD double KO mice [33,34]. However, no studies have directly tested the contributions of HuD in alternative splicing and polyadenylation at a transcriptome-wide level. Therefore, we sought to examine the effects of HuD deletion on these processes in the neocortex of adult *Elavl4^–/–^* (HuD KO) mice. 

## 2. Results

### 2.1. Alternative Splicing of Transcripts in HuD KO Cortex

There are several forms of alternative splicing mechanisms, including exon skipping, use of mutually exclusive exons, alternative 5′ or 3′ splice sites and intron retention. Each of these events can result in mRNA isoforms with different exons from the same gene or intron-including isoforms of the same gene. Differential splicing events in cortical tissue of HuD KO vs. wildtype (control) littermates were analyzed using the rMATS software. As shown in Figure 1A, 310 significant splicing events were identified. Exon skipping represented the largest proportion of alternative splicing between groups at 77.74%, while both intron retention and mutually exclusive exons represented 3.55% (Figure 1B). Complete rMATS output for all five alternative splicing events including all the statistical analyses of the data from replicate samples of the two genotypes is shown in Appendix A. Events with read coverage ≥ 5 (i.e., aligned reads counts greater than 5), ∣Δψ∣ > 5% (change in splicing greater than 5%) and FDR < 0.05 were considered significant (Figure 1A).

Differences in exon or intron inclusion levels between the two mouse genotypes are represented as follows: positive inclusion levels equal greater inclusion in KOs and negative inclusion levels equal decreased inclusion (more exclusion) in KOs. Altogether, we found 144 statistically significant alternative splicing events with increased inclusion levels in KOs and 166 events with decreased inclusion levels in KOs (Figure 1C). Specifically, transcripts from 114 genes exhibited increased exon inclusion in KOs, also indicating a decrease in exon skipping events. Exon inclusion levels in mRNAs from 127 genes were decreased in the KOs, indicating an increase in exon skipping events. 

Ingenuity Pathway Analysis (IPA) software was used to determine the biological systems impacted by differential splicing events in HuD KOs (Figure 1D,E). The most affected biological pathways concerned cell death and survival, neurological disease, organismal injury and abnormalities, and nervous system development and function (Figure 1D). Examples of major neuronal functions associated with those categories include loss (*p*-value = 0.0273) and viability (*p* = 0.00204) of neurons and synaptic transmission of nervous tissue and pyramidal neurons (*p* = 0.0152 and *p* = 0.0406, respectively) (Figure 1E). 

Since HuD KO was found to have the largest effect on exon skipping, we examined these events in more detail. To identify exons involved in each event, rMATS and Maser outputs were used to determine chromosomal locations of exon start sites in Integrative Genomics Viewer (IGV; Broad Institute, Cambridge, MA, USA). Alternative splicing was then visualized using the rmats2sashimiplot software. Not surprisingly, the exon skipping event with the greatest inclusion level difference occurred in exon 2 of the *Elavl4* gene itself, which is the exon deleted in HuD KO mice. While control mice had 100% inclusion of exon 2 (Figure 2A), HuD KO cortex exhibited 0% inclusion, and reads were shifted to the intron immediately following the exon (Figure 2B). The inclusion levels and statistical analysis of this and other significantly skipped exons are shown in Appendix A.

Other genes where exon skipping was greatly impacted by HuD KO were *Ap4e1* and *Rapgef4*. In this case, both genes were found to have a positive inclusion level difference, indicating that skipping of exon 10 in *Ap4e1* and exon 7 in *Rapgef4* occurs less frequently in KOs than in controls (Figure 2C,D). Alternative splicing of *Ap4e1* at exon 10 has not been reported before, so this may be a novel isoform. The gene encodes the AP-4 complex subunit epsilon-1, which is involved in intracellular trafficking and sorting of AMPA receptors to axons [35]. *Rapgef4* encodes the exchange protein directly activated by cAMP 2 (Epac2), which has been shown to regulate the release of excitatory neurotransmitters [36]. Alternative splicing of exon 7 in *Rapgef4* has been reported previously, with Epac2A being the major splice variant expressed in the brain [37]. 

Although HuD KO significantly affected alternative splicing of several genes, it was unclear whether this was a direct effect of HuD or a result of indirect compensatory mechanisms in the KO mice. To identify genes that were directly affected by KO of this protein, we focused only on transcripts that had been shown to directly bind to HuD by RNA immunoprecipitation (RIP) assays. Significant splicing events were then compared with our previously identified 738 HuD targets (Appendix A). These included common HuD targets from RIP-Chip and GST-HuD pulldowns of mouse forebrain [38] and RIP-seq from mouse striatum (Gardiner et al., manuscript in preparation). Comparison of this list with significant rMATS events (Appendix A) identified 17 genes with transcripts known to bind to HuD that also displayed alternative splicing changes: *Cspp1, Whsc1l1, Atp2c1, Whsc1l1, Ap3s1, Cbx3, Sbno1, Per3, Gria2, Atp2c1, Derl2, Ttc3, Clint1, Ube2w, Ube2w, Snap25, Stau2, Snap23, Ptpn12, Dram2, Ube2w* and *Cbx3*. Multiple alternative splicing events were found in some of these genes (*Atp2c1, Cbx3, Ube2w* and *Whsc1l1*), and the majority of events were found to be exon skipping (Figure 3A). 

The gene that exhibited the greatest inclusion level difference (41.3%) in KOs relative to controls was Cbx3, which encodes chromobox 3 (CBX3), a protein involved in transcriptional repression through the binding of histone H3 tails at methylated sites [39]. The highest inclusion level difference occurred at exon 3. However, we found that the downstream exon for this event was exon 5, indicating that exon 4 is coregulated with exon 3 (Figure 3B). In contrast, the overall read counts of exon 3 were greater in control mice (Figure 3B), stressing the importance of using appropriate methods for identifying alternative splicing events instead of individual exon reads. The lowest inclusion level difference in KOs relative to controls (−39.1%) was found in *Cspp1.* This gene encodes the centrosome and spindle pole associated protein 1 (CSPP1), which functions in spindle organization and is required for primary cilia formation [40,41]. Primary cilia are known to be critical for neuronal development [42]. Alternative splicing of exon 17 has been documented in *Cspp1*, resulting in a long isoform that is more physiologically relevant during mitosis [43]. In contrast, we found an exon skipping event at exon 12, which is excluded more frequently in HuD KOs than in controls (Figure 3C). In this case, read coverage also indicated greater number of reads in controls relative to KOs.

From the 17 genes with mRNAs that were alternatively spliced and targeted by HuD, splicing has been shown to functionally impact two genes, *Snap25* and *Gria2*, which encode proteins primarily involved in synaptic transmission and plasticity. In HuD KO cortices, there was a significant increase in *Snap25* exon 5 inclusion (Figure 4A). 

Furthermore, there are two *Snap25* exon 5 isoforms: exon 5a and exon 5b (Figure 4B). SNAP-25 a and b isoforms differ in their ability to promote vesicle priming and release, with the SNAP-25b isoform primarily expressed in mature neurons [44,45]. In this mRNA, skipping of exon 5b was decreased in HuD KOs compared to controls (Figure 4C). Although the overall reduction of this exon was 8%, since we used bulk RNA-seq for the analyses, it is possible that only a low percentage of neurons was affected by this change.

For *Gria2,* inclusion of exon 14 was decreased in HuD KOs, indicating exon skipping occurred more frequently in these mice (Figure 4D). This gene encodes the glutamate receptor 2 (GluR2) protein, an AMPA receptor subunit involved in excitatory neurotransmission [46]. Alternative splicing is known to occur between exons 14 and 15, which are specified as “flip” and “flop” exons (Figure 4E). GRIA2 subunits with inclusion of exon 14 are considered the “flop” isoforms, while those with inclusion of exon 15 constitute the “flip” isoforms [47]. Visualization of exons 14 and 15 showed lower read coverage in HuD KOs at exon 14 compared to controls, indicating that KOs contained decreased levels of *Gria2* flop isoforms (Figure 4F). In contrast, there were no changes in alternative splicing at exon 15. Given that the flop GRIA2 shows more rapid AMPA channel opening and faster glutamate desensitization than flip GRIA2 [48,49], our data suggest that HuD may be important in regulating sensitivity of this glutamate receptor through alternative splicing. 

Finally, as shown in Appendix A, none of the alternative splicing differences in HuD KOs resulted in alterations in the overall expression of these genes, including *Ap4e1*, *Rapgef4*, *Cbx3*, *Cspp1*, *Snap25* and *Gria2* (Figure 5). In comparison to this set, 432 genes exhibited significant differences in mRNA levels in HuD KOs (Appendix A).

### 2.2. Alternative Polyadenylation of Transcripts in HuD KO Cortex

To identify alternative polyadenylation events in HuD KOs compared to control mice, we analyzed the RNA-seq datasets using the DaPars software. DaPars reports polyadenylation events through the percent distal usage index (PDUI). The larger the PDUI, the greater the likelihood that the transcript has a lengthened 3′-UTR [50,51]. The program identified significant alternative polyadenylation in 53 genes, with the majority of altered mRNAs shifting towards proximal PAS usage in HuD KOs (Figure 6A). From these, 11 transcripts presented greater PDUI metrics, or lengthened 3′UTRs, in the KOs, and 42 presented smaller PDUI in controls, indicating 3′-UTR shortening in the KOs. Transcripts with lengthened 3′-UTRs and shortened 3′-UTRs are listed in Figure 6B. Complete DaPars outputs including statistical analyses and comparisons with overall changes in gene expression are shown in Appendix A.

Pathway analyses revealed that mRNAs with alternative polyadenylation changes encoded proteins involved in cell-to-cell signaling and interaction, cellular functions, cellular growth, nervous system development and function and neurological disease (Figure 6C). HuD is known to control synaptic plasticity and neurotransmission, so genes involved in these processes were emphasized. Within the “nervous system development and function” category, two of the most prominent genes in KOs that generated transcripts with shortened 3′-UTRs were *Dtnbp1* and *Baiap2* (Figure 6D). *Dtnbp1* encodes dysbindin, or dystrobrevin-binding protein 1, while *Baiap2* encodes brain-specific angiogenesis inhibitor 1-associated protein 2. Using the IPA software, DTNBP1 was found to affect excitation and firing of interneurons (*p*-value = 0.00297 and *p* = 0.00887, respectively), along with the probability of release and size of vesicles (*p* = 0.0322 and *p* = 0.00887, respectively). BAIAP2 was found to be involved in paired-pulse facilitation of collateral synapses *(p* = 0.00297) and the size of postsynaptic densities (*p* = 0.0118). 

To visualize *Dtnbp1* and *Baiap2* 3′-UTRs, read coverage was compared at both DaPars-predicted proximal PAS and inferred distal PAS. Both genes were found to have lower read coverage after the proximal PAS, indicating that HuD KO neurons contained transcripts with shorter 3′-UTRs (Figure 6E,F).

Finally, to test for the possibility of direct effects of HuD on differential polyadenylation events, transcripts from the 53 mRNAs were compared with the previous HuD target dataset (Appendix A). Between these datasets, only three genes in common were identified: *Alg6*, *Max* and *Mmachc* (Figure 7A). 

*Alg6* encodes the dolichyl pyrophosphate Man9GlcNAc2 alpha-1,3-glucosyltransferase enzyme, *Max* encodes the Myc-associated factor X transcription factor and *Mmachc* encodes the methylmalonic aciduria and homocystinuria type C protein. The 3′-UTR of *Alg6* was longer in KOs, while the 3′-UTRs of *Max* and *Mmachc* were shorter in KOs (Figure 7B–D). Despite the changes in 3′-UTR lengths in these HuD target mRNAs, their expression levels did not differ in KOs vs. control cortices (Appendix A). As shown in Figure 8, the overall levels of mRNAs that showed significant alternative polyadenylation did not change in HuD KO cortices. As expected, *Elavl4* mRNA levels were significantly decreased in these tissues (Figure 8). These findings indicate that the observed alternative polyadenylation changes were not the result of differential mRNA stability of the long vs. short 3′-UTR isoforms.

Overall, we found that HuD KO did not affect alternative polyadenylation as much as it did alternative splicing, suggesting that HuD is more involved in regulating splicing in neurons. 

## 3. Discussion

The purpose of this study was to determine transcriptome-wide changes in alternative splicing and alternative polyadenylation mediated by HuD. We found that deletion of HuD preferentially affected splicing of transcripts involved in several neuronal functions, such as synaptic transmission. In agreement with previous findings where exon skipping represents the most frequent alternative splicing event in biology [52], we found that this event also accounted for the largest proportion of alternative splicing events altered in HuD KO cortex.

HuD KO was found to alter splicing of *Rapgef4* and *Ap4e1*, both of which are involved in excitatory neurotransmission. Although the effects of alternative splicing on the function of their proteins are unknown, deletions and variants of these genes have been linked to several nervous system disorders. *Ap4e1* deficiency has been tied to cerebral palsy syndrome, speech deficits and hereditary spastic paraplegia [53,54,55]. Rare variants in *Rapgef2* have also been found in autism patients [56]. Mice deficient in Epac2, or mice expressing the autism-related variant, exhibit deficits in social interactions and altered dendritic morphology [57,58,59]. Interestingly, KO of HuD during development is also known to disrupt dendritic outgrowth [60].

Two genes involved in neurotransmission that were also alternatively spliced and targeted directly by HuD are *Snap25* and *Gria2*. Notably, there is evidence of alternative splicing in neurons occurring at the same exons reported in this study. *Snap25* has two alternatively spliced isoforms, *Snap25a* and *Snap25b*, which are regulated in different stages of development [61,62,63]. The SNAP25-a protein is more abundant in the embryonic mouse brain, while the SNAP25-b protein becomes the predominant form after birth during the major period of synaptogenesis [64,65]. In agreement with our findings of alternative splicing in the *Snap25* transcript, a mutually exclusive exon event at exon 5 was also reported in a double HuC and HuD KO mouse [34].

Flip and flop isoforms of the GluR2 protein encoded by the *Gria2* gene are also differentially expressed in a cell type- and age-specific manner [48]. Alternative splicing of the mRNA occurs in response to activity, mediating short-term plasticity and synapse homeostasis [47,66]. Electrophysiology studies have shown that flop variants of GluR2 desensitize or remain inactivated when glutamate remains bound to the receptor [48]. Flop isoforms also recover more slowly from desensitization compared to flip counterparts [48]. Previous work by our group showed that HuD binds to and regulates the expression of *Gria2* [67]. However, HuD was not associated with alternative splicing of this mRNA until now.

In addition to the neuronal functions and pathways associated with *Snap25* and *Gria2*, both genes have been implicated in several neuropsychiatric disorders. Some studies suggest that *Snap25* imbalances, or changes in expression of SNAP25-a and SNAP25-b isoforms, contribute to ADHD and schizophrenia, as well as alcohol use disorder (AUD) and smoking risk [68,69,70,71]. *Gria2* has also been deemed an addiction-related gene, and HuD has been shown to regulate this mRNA and addiction-related behavior [67]. *Gria2* flop mRNA levels in the orbital frontal cortex have shown a positive correlation with chronic alcohol use [72]. Therefore, it is possible that HuD-mediated alternative splicing of *Snap25* and *Gria2* could play a role in these disorders. 

Two additional HuD targets, *Cbx3* and *Cspp1,* exhibited the greatest alternative splicing effect size in HuD KOs. CBX3, also known as HP1γ, is a chromatin protein that regulates the transcriptional response through gene silencing in neuronal maturation, development and differentiation [73,74,75]. Additionally, somatic deletions in the gene were found in the cerebellum of schizophrenia patients [76,77]. *Cspp1* is broadly expressed in neurons, and the CSPP1 protein is thought to be involved in neural-specific functions of primary cilia [78,79]. Mutations in *Cspp1* have also been linked to Joubert syndrome, a developmental brain disorder [78]. Still, alternative splicing of exons 3 and 4 in *Cbx3* and exon 12 in *Cspp1* have not been reported before.

In addition to alternative splicing, HuD KO neurons also exhibited changes in alternative polyadenylation. There was a higher proportion of transcripts with shorter 3′-UTRs in HuD KO cortices. HuD is known to bind the proximal PAS and physically block cleavage at the site; thus, lack of HuD binding at the proximal PAS would result in shorter 3′-UTRs [80,81]. Supporting this idea, we previously showed that overexpression of HuD in the dentate gyrus increases the levels of the long 3′-UTR transcripts of BDNF [82]. Since longer BDNF mRNA isoforms localize to dendrites and axons, deletion of HuD could impact dendritic and axonal transcriptomes, as well as overall gene expression profiles of mRNAs with altered alternative polyadenylation.

Within the set of transcripts with significantly shorter 3′-UTRs in HuD KOs, *Baiap2* and *Dtnbp1* are involved in neurotransmission. The BAIAP2 protein is primarily implicated in actin dynamics at the postsynaptic density of excitatory synapses [83]. There are four isoforms of the protein and three validated PAS; however, the functions of these isoforms remain unknown [84]. DTNBP1 modulates AMPAR-mediated synaptic transmission and plasticity [85]. It is considered a schizophrenia susceptibility gene, and its expression is reduced in the prefrontal cortex and midbrain of patients with this disorder [86]. Furthermore, variations in the 3′-UTR of *Dtnbp1* have been found to alter the expression of the gene [86]. DTNBP1 variants in the prefrontal cortex are thought to contribute to the pathophysiology of schizophrenia through alterations in glutamatergic transmission [87,88]. HuD transcript levels have also been found to be increased in the prefrontal cortex of schizophrenia patients [89]. Consequently, it is possible that HuD-mediated alternative polyadenylation of *Dtnbp1* alters excitatory signaling in these patients.

Comparison of direct cytoplasmic HuD targets and alternatively polyadenylated mRNAs revealed only three transcripts: *Alg6*, *Max* and *Mmachc*. Patients with inherited disorders involving *Alg6* display epilepsy, stroke-like episodes, developmental delay, neuropathy and structural abnormalities [90,91]. The MAX protein itself, or in heterodimers with MAD, has been shown to repress c-Myc targets and promote neuronal differentiation [92]. Patients with mutations in the *Mmachc* gene exhibit cognitive impairment, epilepsy, ataxia, and pyramidal and peripheral nerve symptoms [93]. In the case of the *Max* transcript, the predicted PAS was identified in an intron, which is only known to occur in approximately 20% of human genes [94]. Furthermore, this site is listed in the new PolyASite 2.0 database [95] and was found to occur in neurons as shown by a single-cell RNA-seq study in mouse tissues [96].

Like patients with mutations in genes associated with altered polyadenylation, HuD KO mice show several nervous system abnormalities, including defective dendritogenesis and altered neuron maturation in the neocortex and hippocampus [60,97]. Additionally, HuD KO mice have shown behavioral deficits, such as an abnormal clasping reflex [97] and difficulties in learning and memory [60]. Although no electrophysiological studies have been performed in HuD KOs, we previously showed that overexpression of this protein in hippocampal mossy fiber projections to CA3 neurons increases paired-pulse facilitation [98]. By comparison, it is likely that KO mice would show deficits in glutamatergic transmission at the same synapses. 

In conclusion, our findings suggest that alternative splicing and polyadenylation events in genes involved in neuronal development and synaptic transmission could contribute to the structural and functional deficits observed in HuD KO mice.

## 4. Materials and Methods

### 4.1. Animals

HuD KO (*Elavl4*^–/–^) mice were a gift from Prof. Hideyuki Okano. These mice were created by removing exon 2 of the mouse Elavl4 gene [97]. Briefly, exon 2 of the Elavl4 transcript was replaced with a PGK-Neo vector, causing a frameshift mutation. As controls, we used wildtype littermates from the same line. Mice were backcrossed to C57BL/6 for at least ten generations. 

### 4.2. RNA sequencing (RNA-seq)

Cortices were dissected from three adult male HuD KO mice and three wildtype (control) littermates of approximately 3–4 months of age. Total RNA was extracted using Trizol (Invitrogen, Carlsbad, CA, USA) and quantified using Qubit (Bio-Rad, Hercules, CA, USA). RNA quality was determined using both agarose gels and NanoDrop 1000 (Thermo Fisher Scientific, Waltham, MA, USA) absorbances. Aliquots of 2 µg RNA from three mice of each genotype were sent to Arraystar Inc. (Rockville, MD, USA) for paired-end sequencing using NovaSeq 6000 S4 reagent kits and an Illumina NovaSeq 6000 platform (Illumina, San Diego, CA, USA). Sequencing was carried out by running 150 cycles.

### 4.3. Quality Check of Sequencing Reads and Alignment

Using a custom-written MATLAB code to perform a principal component analysis (PCA) of the RNA-seq data, one HuD KO sequencing pair was considered an outlier and this replicate was excluded from the analysis. An unsupervised heatmap clustering analysis of these individual samples is shown in Appendix A. The quality of raw RNA sequencing reads was evaluated using the FastQC software (version 0.11.5) [99] and adapters were removed using the Cutadapt (version 1.15) and Trimmomatic (version 0.38) software [100,101]. A minimum read length of 145 bp was selected by including the -m parameter in the command and reads greater than 145 bp were trimmed to that length using the CROP parameter in Trimmomatic. RNA-seq data were aligned to the *Mus musculus* genome (UCSC browser, mm10) using STAR (version 2.7.3a; https://github.com/alexdobin/STAR (accessed on 17 March 2021)) [102], and MultiQC (version 1.8) was used to perform a final quality check on STAR alignment files [103]. If alignments were found to be the same read length and have >80% reads mapped to a unique location, the data were considered to be of good quality and alternative splicing and polyadenylation analyses were performed.

### 4.4. Alternative Splicing Analysis 

#### 4.4.1. rMATS

The rMATS software (version 4.0.2) with default settings was used to detect differential alternative splicing events from RNA-seq data [104]. The software was downloaded through the rMATS developer webpage: http://rnaseq-mats.sourceforge.net (accessed on 17 March 2021). BAM files obtained through STAR alignment were input into the program. From the output files, those that calculated differential splicing using reads that mapped to both exons/introns and splice junctions were used. Significant events were filtered using the Maser software package in Program R (version 1.6.0) [105], which was downloaded through the Bioconductor website at https://www.bioconductor.org/packages/release/bioc/html/maser.html (accessed on 17 March 2021).

#### 4.4.2. Visualization of rMATS Results

To visualize rMATS results and differential splicing events between groups, rmats2sashimiplot was used. The tool was downloaded directly from the developer’s GitHub page: https://github.com/Xinglab/rmats2sashimiplot (accessed on 17 March 2021). BAM files and rMATS output files were used to run the program. Coordinates from each event were then input into the IGV software (version 2.8.2) to determine which exon/intron was involved in splicing [106,107].

### 4.5. Alternative Polyadenylation Analysis

#### 4.5.1. BAM to BedGraph Conversion

Before using an alternative polyadenylation software, it was necessary to convert BAM files to BedGraph files. To do so, a chromosome annotation text file for the mm10 genome was downloaded from UCSC. Then, the conversion was performed using the genomecov function of the BedTools software (version 2.26.0) [108], which was downloaded directly from https://github.com/arq5x/bedtools2 (accessed on 17 March 2021). 

#### 4.5.2. DaPars Analysis 

The DaPars software (version 0.9.1) was used to determine alternative polyadenylation between groups [50,51]. The software was downloaded from the developer’s GitHub page: https://github.com/ZhengXia/dapars (accessed on 17 March 2021). Before the use of the tool, a 3′-UTR annotation of the mm10 genome was created using a BED text file and a gene symbol file from the UCSC Table Browser [109]. The DaPars_Extract_Anno.py function was used to create the annotation, and then it was used in the analysis, along with specific parameters for a read coverage cutoff of 30, an FDR cutoff of 0.10, a PDUI cutoff of 0.2 and a fold change cutoff of 0.59. A list of genes involved in alternative polyadenylation was generated as a single output file. Those listed as “Y” passed all the specified parameters and were considered significant.

#### 4.5.3. Visualization of DaPars Results 

To visualize DaPars results, BAM files were analyzed with the IGV software (version 2.8.2) for each group; then, 3′-UTR regions were examined.

### 4.6. Pathway Analysis

Ingenuity Pathway Analysis (IPA) is a bioinformatics tool that utilizes current literature to facilitate the interpretation of gene expression data into biological networks [110]. IPA (QIAGEN Inc., https://www.qiagenbioinformatics.com/products/ingenuity-pathway-analysis (accessed on 17 March 2021)) was used to identify functions, disease implications, and canonical pathways associated with genes identified in alternative splicing and alternative polyadenylation analyses. 

## Figures and Tables

**Figure 1 molecules-26-02836-f001:**
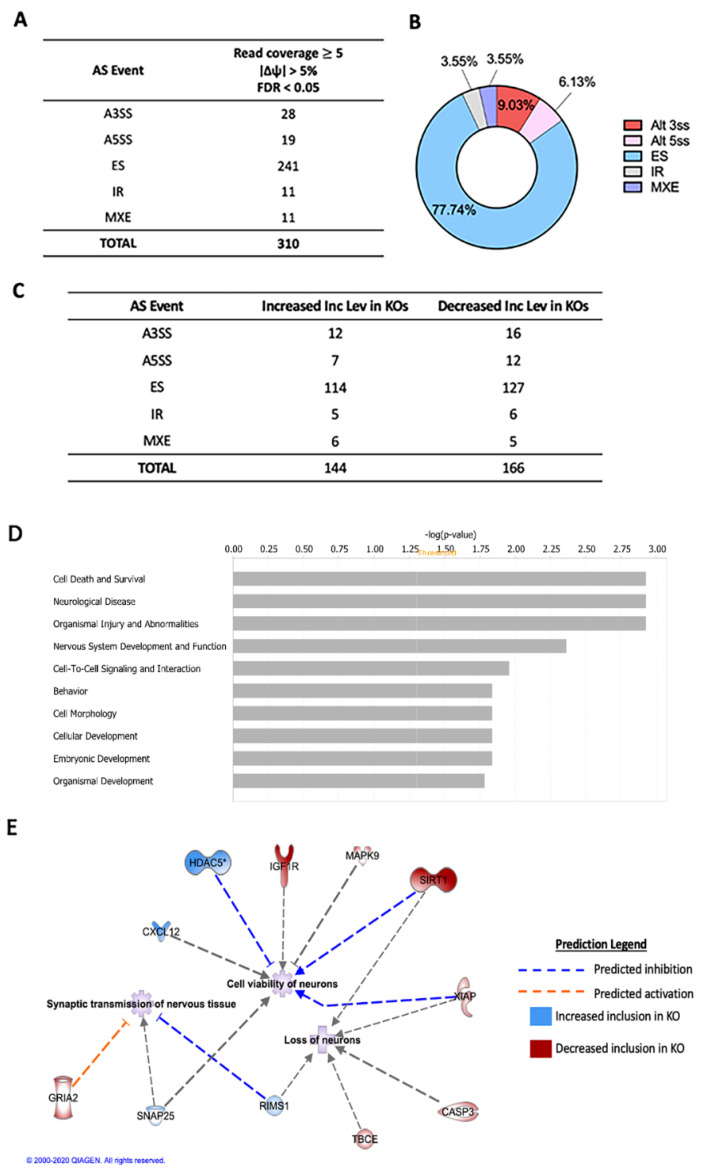
Alternative splicing (AS) events associated with deletion of HuD. (**A**) Total number of significantly different AS events in HuD KO cortices (*n* = 3). (**B**) Proportion of AS differences between KO and controls. (**C**) Number of increased and decreased inclusion AS levels in HuD KOs. (**D**) Top biological pathways associated with AS transcripts in HuD KO cortex analyzed by Ingenuity Pathway Analysis (IPA). Yellow line indicates *p* = 0.05. (**E**) Top neuronal functions affected by alternative splicing of transcripts. Blue lines predict inhibition of the function, while orange lines predict activation. Blue molecules indicate increased exon inclusion in HuD KOs, while red molecules indicate decreased exon inclusion.

**Figure 2 molecules-26-02836-f002:**
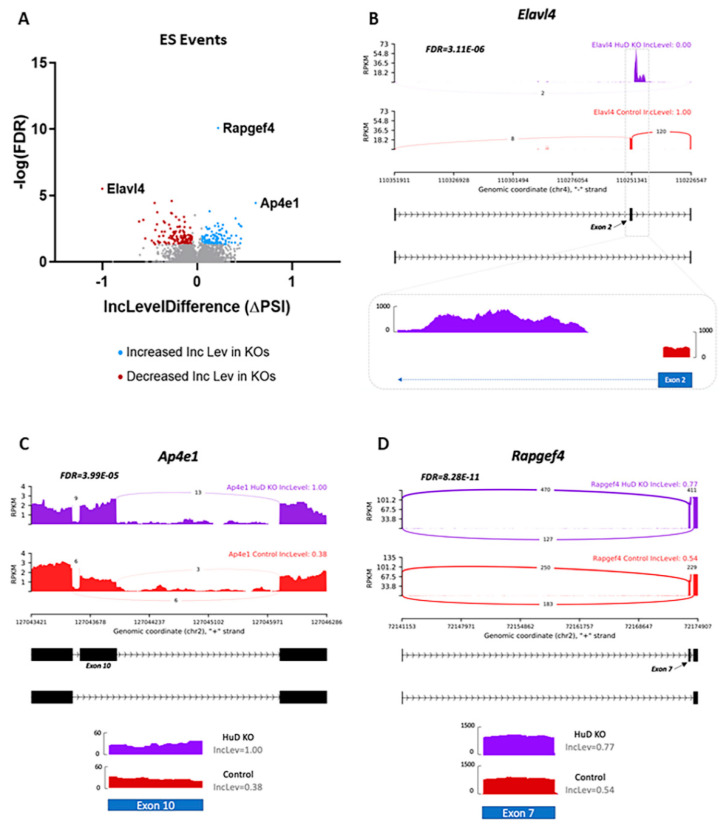
Exon skipping (ES) events associated with HuD KO. (**A**) Volcano plot showing significant changes (−log(FDR)) vs. inclusion level (Inc level) difference (ΔPSI) between HuD KO and control mice. Blue dots show genes with significantly increased inclusion level differences and red dots show those with significantly decreased inclusion levels in HuD KO cortices. The most significant changes are identified by gene name (*n* = 3). (**B**) Top panels show sashimi plots demonstrating exon 2 skipping in the *Elavl4* transcript, which is the exon deleted in HuD KOs. The bottom panel shows read coverage using IGV confirming exon 2 skipping in HuD KO. (**C**) Sashimi plots depicting exon 10 skipping and read coverage for exon 10 in *Ap4e1* and (**D**) sashimi plots and exon 7 coverage in *Rapgef4* in HuD KO and control mice.

**Figure 3 molecules-26-02836-f003:**
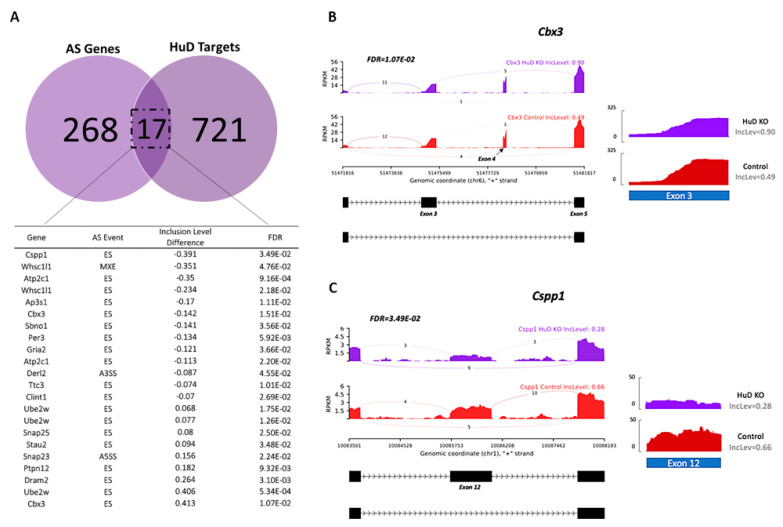
List of transcripts that are alternatively spliced and directly interact with HuD. (**A**) Venn diagram showing the number of transcripts that are HuD targets and alternative spliced in HuD KO. List of genes with exon skipping (ES), alternative 3′ splice sites (A3SS) and alternative 5′ splice sites (A5SS) in HuD KO cortex. A negative inclusion level difference denotes an exon that is more excluded in KOs relative to controls, while a positive value indicates an exon with greater inclusion in HuD KO (*n* = 3). (**B**) Sashimi plots and read coverage of exon 3 in the *Cbx3* transcript. This exon is the top included exon in KO mice. (**C**) Sashimi plots and read coverage of exon 12 in the *Cspp1* transcript. This exon is the top excluded exon in KO mice.

**Figure 4 molecules-26-02836-f004:**
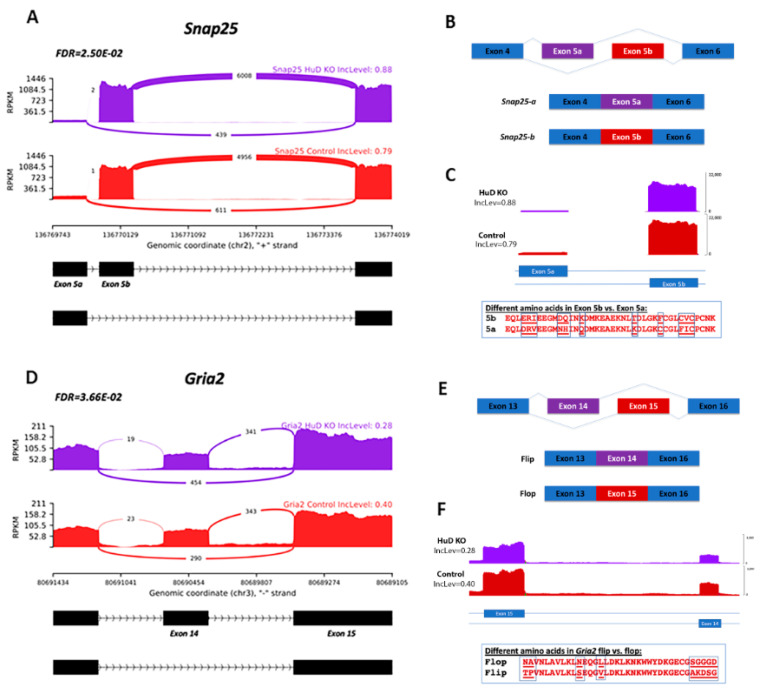
Alternative splicing (AS) of *Snap25*and *Gria2* transcripts in HuD KO cortex. (**A**) *Snap25* sashimi plot depicting decreased exon 5b skipping in HuD KOs (*n* = 3). (**B**) Diagram showing AS of exons 5a and 5b in *Snap25*. (**C**) Read coverage of exon 5b using IGV and amino acid sequence comparison of exons 5a and 5b. (**D**) *Gria2* sashimi plot depicting increased exon 14 skipping in HuD KO cortex (*n* = 3). (**E**) Diagram showing AS of the “flip or flop” isoforms of *Gria2*. (**F**) Read coverage of exon 14 using IGV and amino acid sequence comparison of exons 14 and 15.

**Figure 5 molecules-26-02836-f005:**
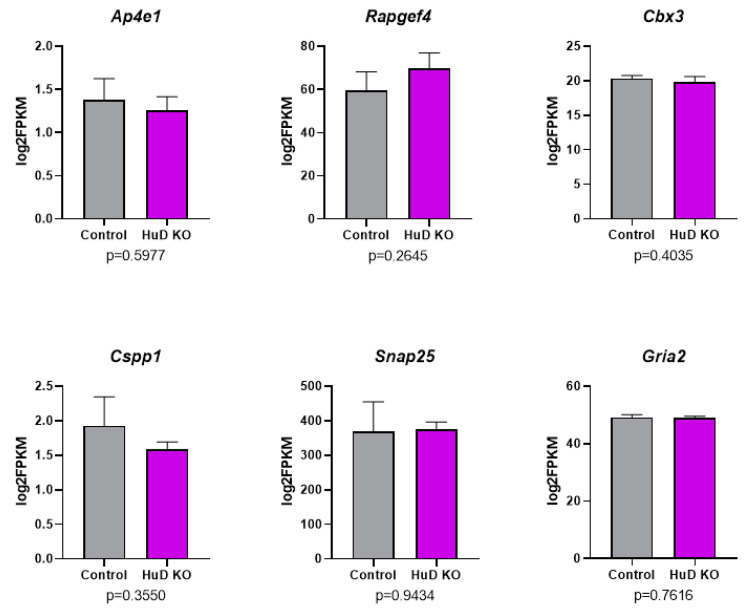
No significant changes in the overall levels of mRNAs that show significant alterations in exon skipping in HuD KO cortex. Panels show the results of RNA-seq levels as log_2_FPKM along with expression level *p*-values for six mRNAs that showed significant changes in exon skipping in HuD KO cortices (*n* = 3).

**Figure 6 molecules-26-02836-f006:**
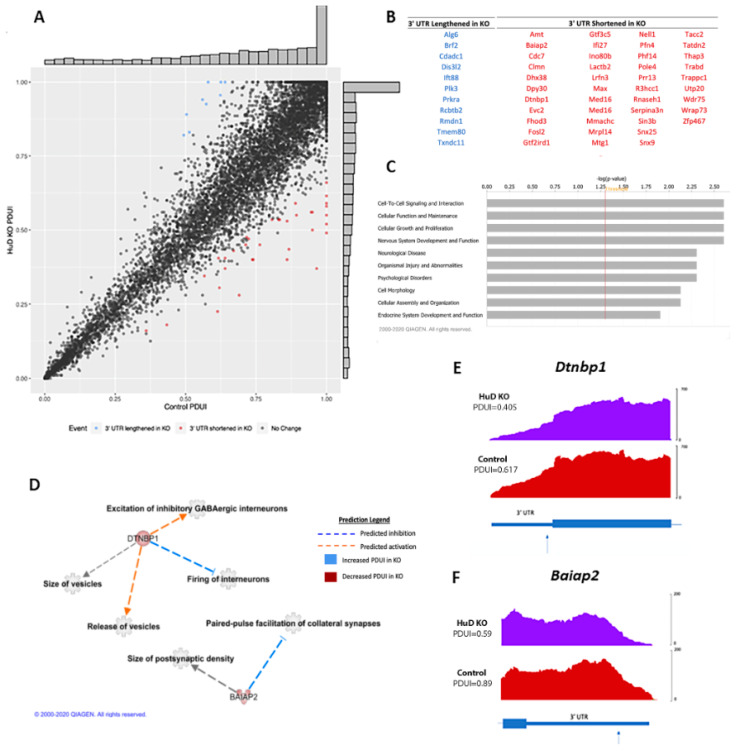
Differences in alternative polyadenylation (APA) of transcripts in HuD KO cortices. (**A**) Scatter plot depicting the percent distal usage index (PDUI) in control and HuD KO mice. Blue points represent significantly lengthened transcripts in HuD KOs, while red points represent significantly shortened transcripts (*n* = 3). (**B**) Complete list of lengthened and shortened transcripts in HuD KO cortex. (**C**) Biological pathways significantly enriched with alternatively polyadenylated transcripts identified by IPA. (**D**) Top neuronal functions affected by APA. Blue lines predict inhibition of the function, while orange lines predict activation. Blue molecules represent increased PDUI in HuD KOs, while red molecules represent decreased PDUI. (**E**) Read coverage graphs of the 3′-UTRs of *Dtnbp1* and (**F**) *Baiap2*. Both transcripts exhibit decreased PDUI and shorter 3′-UTRs. Arrows show the position of the DaPars predicted proximal poly(A) signal (PAS) relative to the last coding exon.

**Figure 7 molecules-26-02836-f007:**
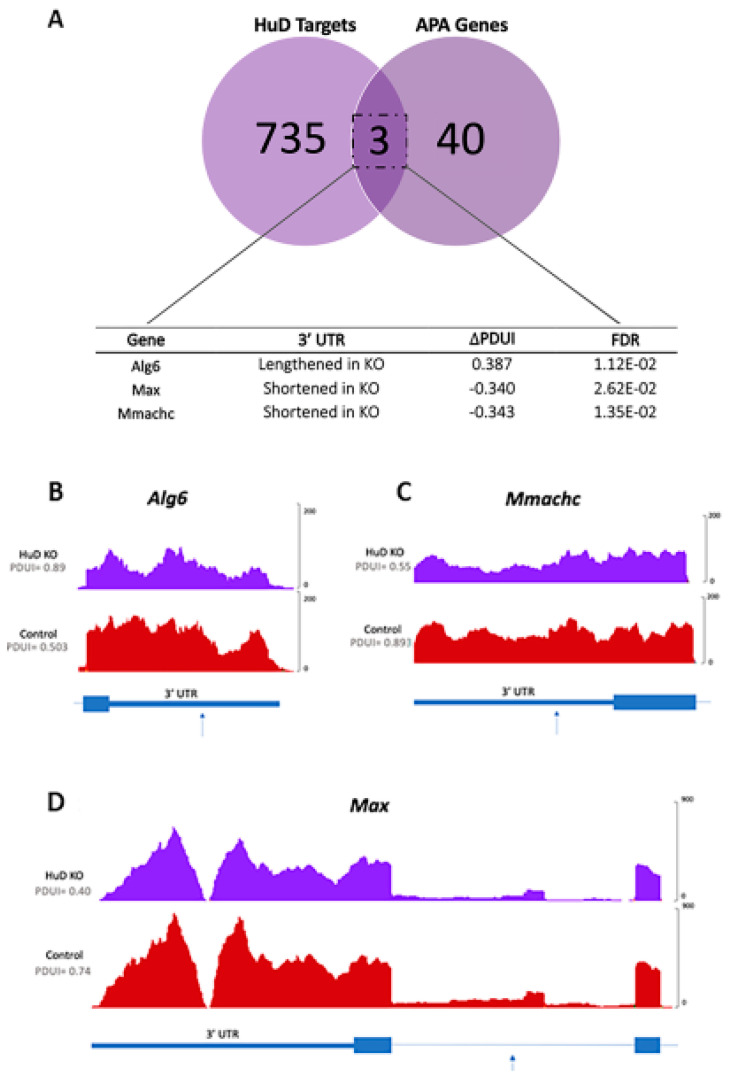
Alternative polyadenylation of transcripts that directly interact with HuD. (**A**) Venn diagram and list of HuD targets that are alternatively polyadenylated in HuD KO mice. The 3′-UTR of *Alg6* is lengthened in HuD KO cortex, while the 3′-UTRs of *Max* and *Mmachc* are shortened (*n =* 3). (**B**–**D**) Read coverage graphs of *Alg6* 3′-UTR, (**B**) *Mmachc* 3′-UTR (**C**) and *Max* 3′-UTR (**D**). Arrows indicate the position of the DaPars-predicted proximal PAS. For the *Max* transcript, the proximal PAS occurs in an intron.

**Figure 8 molecules-26-02836-f008:**
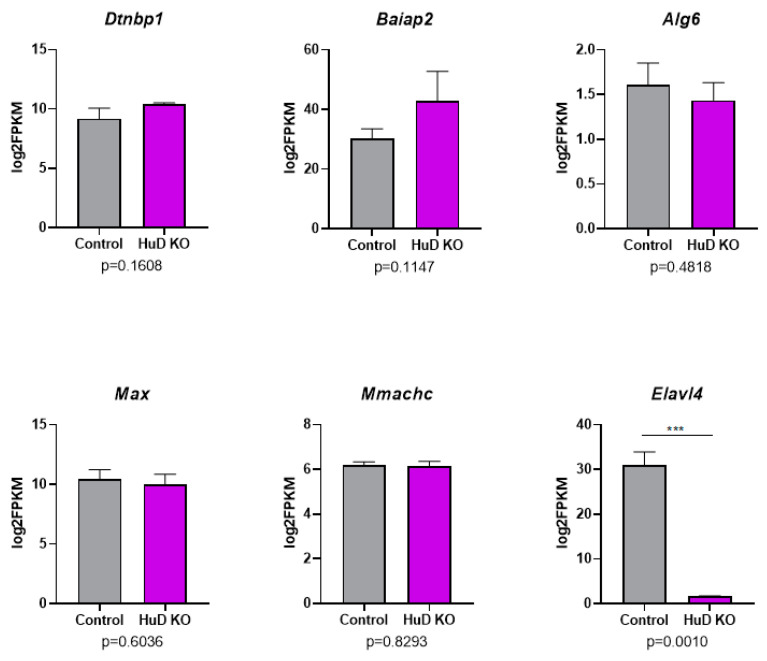
No significant changes in the overall levels of mRNAs that showed significant alternative polyadenylation in HuD KO cortices. Panels show the results of RNA-seq levels as log_2_FPKM along with *p*-values for five mRNAs that showed significant changes in alternative polyadenylation in HuD KO mice (*n* = 3). *** The significant decreases in *Elavl4* mRNA levels in HuD KO mice are shown as a comparison set (*n* = 3).

## Data Availability

RNA sequencing data are available at NCBI Gene Expression Omni.

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
