# Peer review of "The RNA-Binding Protein HuD Regulates Alternative Splicing and Alternative Polyadenylation in the Mouse Neocortex"

_molecules, 2021, doi:10.3390/molecules26102836_

Round 1
Reviewer 1 Report
This manuscript provides a solely computational analysis of transcriptome-wide effects of HuD deletion using RNA sequencing data obtained from the neocortex of Elavl4-/- mice generated by Ogawa et al. They investigated both alternative splicing data and found 310 genes regulated genes, and altered PAS usage in 53 genes. The authors integrated RIP-Chip and GST-HuD pulldown data of mouse forebrain and RIP-seq data from mouse striatum to distinguish between direct targets and secondary effects on HuD depletion. The authors provide their results in form of tables which will be useful resource for future comparative studies. Encouragingly, their analysis retrieved 17 validated HuD targets such as Cbx3, Cspp1, Snap25 and Gria2. Many of their AS events involve interesting and neurologically relevant genes.
For both their AS and altPAS analysis the authors provide individual sequencing tracts as evidence of functionality. The authors should provide quantification of replicates of the RNAseq data and statistical analysis of their events. To make any claims on transcriptomic changes these have to be reproducible between replicates and significantly different from the control condition.
At the moment it is difficult to distinguish if an overall stabilisation of the transcript or an actually change in PAS usage as visualised for selected example of the gene Max. The authors should provide a more convincing analysis to distinguish these possibilities, ideally by some qPCR based validations.
It is not clear how the authors defined their PAS data set. Did they use annotated PASs as defined by the bioinformatics algorithm DaPars or did they use a more up-to-date list. They might be able to extend their list by using a more present PAS annotation data set like for example provided by Herrmann et. al. in NAR 2020.
Also, using conventional RNAseq to investigate PAS switches will be hampered by the difficulty analysing changes particularly of shorter isoforms, especially if they are the minor variant. The authors could consider using 3’ RNAseq methods to instigate these questions in more depth. As 3’ RNAseq coverage is limited only to parts of 3’UTRs this greatly reduces the sequencing depth requirements to power PAS analysis and consequently costs to generate a more meaningful data set. This is only a recommendation for future studies as this request would go beyond the scope of this manuscript.
Finally, just a minor comment on the figures: The labelling is inconsistent and the font size for most of the graph is too small. This should be adjusted.
Author Response
Reviewer 1:
The authors provide their results in form of tables which will be useful resource for future comparative studies. Encouragingly, their analysis retrieved 17 validated HuD targets such as Cbx3, Cspp1, Snap25 and Gria2. Many of their AS events involve interesting and neurologically relevant genes.
For both their AS and altPAS analysis the authors provide individual sequencing tracts as evidence of functionality.
- The authors should provide quantification of replicates of the RNAseq data and statistical analysis of their events. To make any claims on transcriptomic changes these have to be reproducible between replicates and significantly different from the control condition.
Reply 1: We apologize for the confusion but we did perform statistical analyses of replicates data of the two genotypes as shown in the Supplementary tables. Also, the statistical analyses were shown in Figures 1A and 6A.
To further clarify this, we added the following sentence in the Results:
“Complete rMATS output for all five alternative splicing events including all the statistical analyses of the data from replicate samples of the two genotypes as are shown in Tables S1-S5. Events with read coverage ≥ 5 (i.e., aligned reads counts greater than 5), ∣Δψ∣ > 5% (change in splicing greater than 5%), and FDR < 0.05 were considered significant.”
- At the moment it is difficult to distinguish if an overall stabilisation of the transcript or an actually change in PAS usage as visualised for selected example of the gene Max. The authors should provide a more convincing analysis to distinguish these possibilities, ideally by some qPCR based validations.
Reply 2: In Table S6, we provided data on the expression levels of Max as well as other alternatively polyadenylated mRNAs and none of these have any changes in expression levels. These results are consistent with a role of HuD in alternative polyadenylation rather than alternative stability of the short vs. long isoforms of the mRNAs. To make this clear we have added the following sentence in the Results:
“Despite the changes in 3’UTR lengths in these HuD target mRNAs, their expression levels did not differ in KOs vs. control cortices (Table S7). These findings indicate that the observed alternative polyadenylation changes are not the result of differential mRNA stability of the long vs. short 3’UTR isoforms.”
- It is not clear how the authors defined their PAS data set. Did they use annotated PASs as defined by the bioinformatics algorithm DaPars or did they use a more up-to-date list. They might be able to extend their list by using a more present PAS annotation data set like for example provided by Herrmann et. al. in NAR 2020.
Reply 3: Although we used the DaPars annotate PASs, we were able to validate our results using the PolyAsite2.0 databased described in Herrmann et al, NAR 2020. As an example, we indicated in the text that the alternatively polyadenylation site in the intron of Max was also identified in the PoAsite2.0 database and that it occurs in neurons as shown by an article of the Tabula muris consortium that we also cited as shown below:
“ In the case of the Max transcript, the predicted PAS was identified in an intron, which is only known to occur in approximately 20% of human genes [94]. Furthermore, this site is listed in the new PolyASite 2.0 database [95] and was found to occur in neurons as shown by a single-cell RNA-seq study in mouse tissues [96].”
- Finally, just a minor comment on the figures: The labelling is inconsistent and the font size for most of the graph is too small. This should be adjusted.
Reply 4: We agree with the reviewer and increased the font size in most of the figures. The different in the labeling of the panels is due to the different configuration of the figures that are either vertical or horizontal. We were not aware of the format of articles in the journal.
Reviewer 2 Report
This paper describes the results of RNA sequence-level analysis of mouse HuD function, and is very interesting. It contains the results of various experiments and is expected to be of interest to many readers.
On the other hand, it is necessary to provide sufficient explanations and data on whether the individual differences in the experimental animals were taken into account in the data submitted. Using the minimum number of animal populations required, an examination of the variability among individuals and the results of appropriate statistical tests are needed.
Author Response
Reviewer 2:
On the other hand, it is necessary to provide sufficient explanations and data on whether the individual differences in the experimental animals were taken into account in the data submitted. Using the minimum number of animal populations required, an examination of the variability among individuals and the results of appropriate statistical tests are needed.
Reply: We apologize for the confusion. As described in reply to Rev 1 point 1, we did perform statistical analyses of all the data as shown in the Supplementary tables. Also, statistical analyses were shown in Figures 1A and 6A.
To further clarify this, we added the following sentence in the Results:
“Complete rMATS output for all five alternative splicing events including all the statistical analyses of the data from replicate samples of the two genotypes as are shown in Tables S1-S5. Events with read coverage ≥ 5 (i.e., aligned reads counts greater than 5), ∣Δψ∣ > 5% (change in splicing greater than 5%), and FDR < 0.05 were considered significant.”
Finally, an unsupervised heatmap clustering analysis of replicate samples is now shown in Figure S1.
Round 2
Reviewer 1 Report
I thanking the author for the clarifications and additions to the manuscript.
I still think it would aid the manuscript to add the graphs showing gene expression level, with proper statistics shown and p-Values/FDR added. Therefore I would appreciate – as will the future readers of the manuscript - to include all the gene expression levels and all the statistical tests in the main figures/Figure legend for the presented genes and only include genes with significantly different levels. Some of the events like Snap25 appear to have a 1% FC which I struggle to believe will have biological significance.
The tables are an useful addition, but it would help if they were sorted by FC and/or p-value and not alphabetically.
Do the FC levels correspond to the IncLevelDifferences (delta PSI) from Figure 2A?
Author Response
Reply to each point of reviewer 1:
1) I still think it would aid the manuscript to add the graphs showing gene expression level, with proper statistics shown and p-Values/FDR added. Therefore, I would appreciate – as will the future readers of the manuscript - to include all the gene expression levels and all the statistical tests in the main figures/Figure legend for the presented genes and only include genes with significantly different levels.
Reply 1: We thank the reviewer for this important recommendation. We now indicate both in the text and Tables S1-S5 that none of the alternative splicing differences in HuD KOs resulted in alterations in the overall expression of these genes, including Ap4e1, Rapgef4, Cbx3, Cspp1, Snap25 and Gria2, which are now shown in a new Figure 5. We also included a new Table S7 to list the 432 genes with significant differences in mRNA levels in HuD KOs.
2) Some of the events like Snap25 appear to have a 1% FC which I struggle to believe will have biological significance.
Reply 2: The deltaPSI for the Snap 25 5b exon was 8% not 1%. To clarify this and propose a potential mechanism we included the following sentence in the results: “Although the overall reduction of this exon was 8%, since we used bulk RNA-seq for the analyses, it is possible that only a low percentage of neurons were affected by this change”
3) The tables are useful additions, but it would help if they were sorted by FC and/or p-value and not alphabetically.
Reply 3:Thanks again for the suggestion. We are now sorting the data on the tables first by FC and then by p-values.
4) Do the FC levels correspond to the IncLevelDifferences (delta PSI) from Figure 2A?
Reply 4: No, the Y axis shows the levels of significance expressed as -log(FDR) vs. IncLevelDifferences (delta PSI) in the x axis. There are no changes in expression levels in any of these genes as shown in Tables S1-S5 and the new Figure 5. To make this point clearer, we now indicate in the legend to Figure 2 that panel A shows a “Volcano plot showing significant changes [-log(FDR)] in inclusion levels difference (ΔPSI) between HuD KO vs control cortices.
Reviewer 2 Report
Please show the numbers of animal used in each figure legend.
Author Response
Please show the numbers of animal used in each figure legend.
Reply: We now included the number of animals in the legends to all the figures.